# Enhancing the Electrochemical Stability of LiNi_0.8_Co_0.1_Mn_0.1_O_2_ Compounds for Lithium-Ion Batteries via Tailoring Precursors Synthesis Temperatures

**DOI:** 10.3390/ma16155398

**Published:** 2023-08-01

**Authors:** Guanhua Zhang, Hao Wang, Zihan Yang, Haoyang Xie, Zhenggang Jia, Yueping Xiong

**Affiliations:** 1Queen Mary University of London Engineering School, Northwestern Polytechnical University, Xi’an 710100, China; 2School of Materials Science and Engineering, Harbin Institute of Technology, Harbin 150001, China; 3School of Chemistry and Chemical Engineering, Harbin Institute of Technology, Harbin 150001, China

**Keywords:** lithium-ion batteries (LIBs), hydrothermal synthesis, electrochemical stability, LiNi_0.8_Co_0.1_Mn_0.1_O_2_ (LNCMO)

## Abstract

LiNi_0.8_Co_0.1_Mn_0.1_O_2_ (LNCMO) cathode materials for lithium-ion batteries (LIBs) were prepared by the hydrothermal synthesis of precursors and high-temperature calcination. The effect of precursor hydrothermal synthesis temperature on the microstructures and electrochemical cycling performances of the Ni-rich LNCMO cathode materials were investigated by SEM, XRD, XPS and electrochemical tests. The results showed that the cathode material prepared using the precursor synthesized at a hydrothermal temperature of 220 °C exhibited the best charge/discharge cycle stability, whose specific capacity retention rate reached 81.94% after 50 cycles. Such enhanced cyclic stability of LNCMO was directly related to the small grain size, high crystallinity and structural stability inherited from the precursor obtained at 220 °C.

## 1. Introduction

Lithium-ion batteries (LIBs) have become promising energy storage devices that are widely used in electric vehicles, portable devices and energy storage systems because of their high energy density, high operating voltage and environmental friendliness [1,2,3]. In recent years, extensive research works have been carried out on layered ternary LiNi_1−x−y_Co_x_Mn_y_O_2_ cathode materials combined with the high structural stability of LiCoO_2_, high capacity of LiNiO_2_ and low cost of LiMnO_2_ [4,5,6]. Among these materials, the Ni-rich LiNi_0.8_Co_0.1_Mn_0.1_O_2_ (LNCMO) compound has attracted much attention since its theoretical capacity is as high as 280 mAh g^−1^. Unfortunately, its high energy barrier inhibits the oxidation of Ni^2+^ to Ni^3+^/Ni^4+^ during high temperature calcination, leading to the existence of some residual Ni^2+^ that occupies the position of Li^+^ as Ni^2+^ and Li^+^ have a similar atomic radius. The residual Ni^2+^ may result in cation mixing and the destruction of its interlayer structure, and thus lead to poor cyclic and rate performance [7,8,9].

The synthesis techniques of LNCMO include coprecipitation [10], sol-gel [11] and hydrothermal [12] and thermal polymerization [13]. Electrode materials prepared by a hydrothermal technique show various advantages such as high purity, tailorable architecture, high preparation efficiency and cost effectiveness [14,15]. Zhang et al. [12] prepared granular LiNi_0.8_Co_0.1_Mn_0.1_O_2_ cathode materials by a hydrothermal technique to inhibit the formation of microcracks. Dahn’s work [16] shows that the metal ions from the incoming salt solution first coordinate to the ammonia present in the solution, and then are slowly released into the basic solution to yield dense spherical hydroxide particles. However, the remaining LiOH on the cathode surface may react with LiPF_6_ in the electrolyte to form HF, resulting in the pulverization of the electrode [17]. It is particularly critical to replace the ammonia source to prepare nickel hydroxide-free precursors. In addition, the unique configuration design also contributes to the improvement of battery performance [18]. The 2D-lamellar stacked cathode designed by Wang et al. [19] showed high-rate capability with 115.8 mA h g^−1^ at 20C, and long-term cyclability.

In this work, LNCMO cathode material was prepared by the synthesis of pine-like precursors via hydrothermal, and then high-temperature, calcination with a lithium source (Li_2_CO_3_). The effect of the precursor hydrothermal temperature on the microstructure, charge/discharge behavior and electrochemical cyclic performance of LNCMO were systematically studied. The results showed that the precursor synthesized at 220 °C exhibited the best electrochemical performance. The obtained LNCMO materials showed enhanced cycling stability and thus may be promising for high-performance LIBs.

## 2. Materials and Methods

### 2.1. Preparation of NCMO Precursors at Various Hydrothermal Temperatures

Pineal Ni_0.8_Co_0.1_Mn_0.1_CO_3_ (NCMO) precursor was prepared by hydrothermal technique using the following steps: (1) Ni(CH_3_COOH)_2_, Co(CH_3_COOH)_2_ and Mn(CH_3_COOH)_2_ were dissolved in deionized water with a molar ratio of 8:1:1, forming a mixture solution with a concentration of 0.25 mol/L; (2) A certain amount of urea, acting as a reaction precipitant, was added to the solution and stirred until the urea was fully dissolved. Then, the solution was transferred to a PPL-lined autoclave for hydrothermal reaction at the temperatures of 160, 180, 200, 220 and 240 °C, respectively, for 12 h; (3) The obtained products were washed with plenty of deionized water and ethanol, and air-dried at 80 °C for 12 h to obtain the precursors.

### 2.2. Preparation of LNCMO Cathode Materials Based on Precursors

LNCMO cathode materials were prepared by high-temperature solid state calcination. The NCMO precursor and Li_2_CO_3_ were thoroughly mixed in the mortar. The mixture was then subjected to a step-wise calcination under air atmosphere at 500 °C for 5 h, and 850 °C for 12 h at a heating rate of 3 °C/min. After cooling in the furnace to ambient temperature, black LNCMO cathode powders were obtained.

### 2.3. Characterization of LNCMO Cathode Materials

The phase composition and crystal structure of NCMO precursors synthesized at various temperatures, and corresponding LNCMO cathode materials, were characterized by X-ray diffraction (XRD, Dmax-rB) with Cu-*Kα* radiation. The morphology was characterized by scanning electron microscope (SEM Merlin Compact). The contents of Ni, Co and Mn were determined by an inductively-coupled plasma atomic emission spectrometer (ICP-AES, Optima 5300DV). The specific surface areas of the LNCMO samples were calculated by the BET equation based on the nitrogen adsorption data on a QUADRASORB-SI/MP-specific surface area and porosity analyzer produced by Kanta Company, USA. X-ray photoelectron spectroscopy (XPS) was carried out to analyze the elemental binding energy using ESCALAB 250Xi of ThermoFisher design.

The electrochemical performance of the as-synthesized LNCMO cathode material was carried out on coin-shape cells using metallic lithium foil as a counter electrode, Celgard 2400 as a separator, and 1 M LiPF_6_ in 1:1 (volume) ethylene carbonate: ethyl methyl carbonate as an electrolyte. The working electrodes were prepared by mixing 80 wt.% active materials, 10 wt.% conductive carbon Super P, and 10 wt.% polyvinylidene fluoride dissolved in N-methyl pyrrolidinone. The resultant slurry was spread uniformly on an aluminum foil and dried at 80 °C under vacuum for 10 h. The pole pieces were cut according to the specifications of the battery case using a skiving machine. The whole cell assembly process was conducted in a glove box filled with highly pure argon. The charge/discharge cycling test was carried out over the potential range of 3.0–4.3 V using a NEWARE BST-5V10mA computer-controlled battery test station at the rate of 20 mA g^−1^.

## 3. Results and Discussion

### 3.1. NCMO Precursors Synthesized at Various Temperatures

In order to investigate the effect of hydrothermal temperature on battery performance, we first characterized the structure of the precursor obtained after hydrothermal treatment. Figure 1 shows the XRD patterns of the NCMO precursors prepared at different hydrothermal temperatures. The XRD patterns of the precursors prepared at all temperatures consist only of the diffraction peaks of NiCO_3_, without other detectable phases. To analyze the composition of precursors prepared at different hydrothermal temperatures, the ICP-AES test was carried out, as listed in Table 1. It shows that the precursor contains three transition metal elements, i.e., Ni, Co and Mn, indicating that the precursor is a ternary solid solution structure with Co and Mn doped in NiCO_3_. The molar ratios of Ni, Co and Mn in the precursors at hydrothermal temperatures of 160, 180, 200, 220 and 240 °C are 4.50:0.90:1, 8.51:1.22:1, 8.30:1.20:1, 8.29:1.21:1 and 8.01:1.02:1, respectively.

Furthermore, the preparation process of LNCMO is a two-step method that consists of hydrothermal (to obtain the precursor) and mixed calcination. We control the ratio of nickel, cobalt and manganese in the raw material at 8:1:1. The precursors obtained at different hydrothermal temperatures show different element contents from the raw material. If the driving force of the reaction is lower than the activation energy, the concentration of activated atoms will be insufficient, resulting in a decrease in the content of the reaction product [20]. At the hydrothermal temperature of 160 °C, the driving force cannot meet the activation energy required for the formation of NiCO_3_, resulting in low crystallinity of the precursor and relatively low Ni content. In addition, Sun et al. [16] found that different metal ions have different complexation rates (i.e., Ni^2+^ > Co^2+^ > Mn^2+^) with ammonia during the hydrothermal process. Therefore, at a higher hydrothermal temperature (180 °C), the content of Co in the precursor is much higher than that of Mn. On the other hand, when the hydrothermal temperature reaches 240 °C, the reaction driving force is high enough to activate excess metal ions [21], resulting in rapid nucleation and growth of all three metal cations in situ. Therefore, the ratio of metal elements in the precursor is close to that in the raw material.

The morphology of the precursors synthesized at different hydrothermal temperatures was observed by SEM. The typical images are shown in Figure 2. All of the synthesized precursors show pine cone shapes formed by the accumulation of flaky primary particles. It is also noted that, as the hydrothermal temperature increases from 160 to 180 °C, the size of the particles decreases; then keeps almost intact at 200 and 220 °C; and finally increases again at 240 °C. The change of the particle sizes may be explained by the crystal nucleation and growth of the precursors. According to relevant literatures [16,22] and XRD results, the reaction equation during the hydrothermal process can be described as:CO(NH_2_)_2_ + H_2_O → 2NH_3_ + CO_2_(1)
M^2+^ + nNH_3_ → [M(NH_3_)_n_]^2+^(2)
[M(NH_3_)_n_]^2+^ + CO_2_ + H_2_O → MCO_3_↓ + nNH_3_(3)

According to Equations (1)–(3), the synthesis of precursors consists of the following steps: (1) urea hydrolysis produces ammonia NH_3_, (2) NH_3_ and metal ion (M^2+^) complexes combine to form [M(NH_3_)_n_]^2+^ nanosheets, forming early-stage primary precursor crystals, and then (3) secondary crystals undergo secondary self-assembly to form MCO_3_ with the presence of CO_2_ and H_2_O. In these processes, self-assembly can be fully performed when the reaction time is long enough. It can also be noticed from Figure 2 that the crystals show spiral growth characters, which are derived from the energy of different crystalline interfaces depending on crystal orientation and primary crystal grains.

In the reaction process, urea act as a complexing agent that induces the formation of precipitate. In addition, urea can reduce the saturation of the solution and lower the nucleation rate [12]. As a result, the crystal nucleation and growth rates can reach certain equilibrium states and thereby induce the special pine cone shape. The smaller the secondary particle size in the pine-like structure, the larger the specific surface area, which is beneficial to the conductivity and energy density of the positive electrode materials. The secondary grain size (long path) of NCMO in Figure 2 synthesized at 160, 180, 200, 220 and 220 °C is 6.6 ± 0.3, 4.5 ± 0.2, 4.3 ± 0.2, 4.1 ± 0.4 and 6.4 ± 0.3 μm, respectively. In addition, the variation of crystallite size is similar to that of particle size. The obtained smaller crystallite size is conducive to shortening the migration path of lithium-ions in the positive electrode material and higher lithium-ion conductivity [23]. When the hydrothermal temperature is 160 °C, the driving force is too low for the formation of NiCO_3_, resulting in the low crystallinity of the precursor and relatively low Ni content. With the increase of hydrothermal temperature, more atoms may participate in the reaction process, which is conducive to the nucleation and growth of grains and the improvement of crystallinity. When the hydrothermal temperature increases to 240 °C, the excessive reaction driving force leads to an extremely fast reaction rate. Such a high rapid reaction rate is not conducive to the growth of single crystals [24], and thus eventually leads to the decline of crystallinity. By comparing the XRD half-peak width of the precursor, it is confirmed that that maximum crystallinity is obtained at 220 °C.

The precursor with the special pine core structure has a large specific surface area. Figure 3 shows the N_2_ adsorption desorption and pore size distribution curves of the precursor prepared at 180 °C and 220 °C. The specific surface areas of the precursors synthesized at 180 °C and 220 °C were calculated by BET formula as 2.80 m^2^ g^−1^ and 2.94 m^2^ g^−1^, respectively. A loop line was formed during the adsorption and desorption processes, indicating the presence of pores on the sample surface. In the pore size distribution curve (inset in Figure 3), there is a clear peak at 40 nm, indicating that the precursor is a mesoporous structure. Such a mesoporous structure not only provides a sufficient surface area for the even distribution of lithium salts on the precursor, but also prevents particle aggregation.

The structure and pore size of precursors play a crucial role in the preparation of high-performance positive electrode materials. The pine cone-like morphology, with smaller particle size, larger specific surface area, and higher vibration density, is conducive to improving the conductivity and energy density of the positive electrode materials [25,26,27]. Although the precursor synthesized at 160 °C has a similar pine cone-like structure, the precursor has poor crystallinity and low nickel atom content. As a result, the precursor synthesized at this temperature will no longer be used to synthesize a positive LNCMO electrode sample.

### 3.2. LNCMO Cathode Materials Based on Precursors Obtained at Different Temperatures

After the high temperature solid state calcination of NCMO precursors synthesized at different hydrothermal temperatures from 180–240 °C, the LNCMO cathode materials were obtained, and their XRD patterns are shown in Figure 4. All of the sharp diffraction peaks can be indexed to the hexagonal α-NaFeO_2_ structure (space group: R3m). The lattice parameters were determined by Bragg equation 2*d*sin*θ* = n*λ*, where *d* is lattice spacing, *θ* is diffraction angle, n is diffraction series and *λ* is wave-length of the irradiation. The corresponding lattice parameters obtained from the XRD patterns are listed in Table 2. The clear splitting of the (006) and (012) peaks and the (108) and (110) peaks indicates a well-ordered, layered structure of all samples [28,29,30,31]. A *c*/*a* value greater than 4.9 also confirms that the material has a good layered structure. The ratio of the peak intensities of (003) and (104) in the four samples is greater than 1.2, indicating low cation mixing of Li/Ni [32]. The peak ratio of (003) and (104) of the sample with a hydrothermal temperature of 220 °C is as high as 1.94, because the precursor synthesized at 220 °C has the highest crystallinity.

SEM images of LNCMO samples obtained from the precursors synthesized at different hydrothermal temperatures are shown in Figure 5. The morphology of LNCMO resembles that of the precursor, i.e., consisting of secondary particles stacked from primary particles. However, different from the precursor, the primary particles of LNCMO show the shapes of blocks or thick sheets, which is attributed to the melting and growth of the primary particles of the precursor during high calcination. In addition, the secondary particle size distribution of samples is uniform, but there are slight differences in the secondary particles, as can be seen from Figure 5. It is noted that the secondary particle size prepared with the precursors of 180 °C and 240 °C is larger, while that of using precursors of 200 °C and 220 °C is smaller, which is basically correlated to the original secondary particle size in the precursor. As a result, the size of the synthesized cathode material is closely related to the original size of the precursor.

The smaller the particle size, the larger the specific surface area; thus, particle size has a direct impact on its electrochemical performance. Positive electrode materials with larger specific surface areas exhibit larger contact areas with the electrolyte, which can improve the conductivity and rate performance of the material. As shown in Figure 5, the secondary grain size (long path) of LNCMO synthesized at 180, 200, 220 and 220 °C is 11.3 ± 0.6, 9.5 ± 0.6, 9.1 ± 0.7 and 12.1 ± 0.6 μm, respectively. In the calcination process, the precursor melting leads to further growth of primary and secondary grains. Combined with XRD analysis, the hydrothermal temperature of 220 °C is considered to be the most favorable for the preparation of the LNCMO cathode.

Previous literatures have shown that the more obvious the splitting of two pairs of peaks (012)/(006) and (108)/(110), the more prominent the layered structure of the LNCMO ternary cathode material. In addition, the higher the peak intensity ratio of *I_(003)_*/*I_(104)_*, the lower the degree of Li^+^/Ni^2+^ mixing, and the more stable is the layered structure. The larger lattice constant *c*/*a* > 4.9 also shows a better layered structure. Figure 6 shows the enlarged images of the two pairs of splitting peaks (012)/(006) and (108)/(110) at diffraction angles 2θ of 37.5–39° and 63.5–65.5°. It is noted that the splitting behavior of both peaks is obvious, indicating that the layered structure of the four samples is relatively complete. In addition, from the lattice constant results of the four samples shown in Table 2, the lattice constant *c*/*a* values of the four samples are >4.9, which also correspond to a good layered structure. On the other hand, the ratio of peak strengths (003) and (104) of the four samples is >1.2, indicating little cation mixing behavior in the samples. Overall, precursors with hydrothermal LNCMO cathode samples prepared with precursors obtained at temperatures between 180–240 °C exhibited good crystallinity with good layered structures.

The XRD analysis results showed that the samples prepared at four hydrothermal temperatures had a certain degree of cation mixing. In order to find a way to avoid cation mixing, the elemental valence states of transition metals in cathode materials were analyzed by XPS. The XPS spectra and peak splitting results of cathode materials prepared at different hydrothermal temperatures are shown in Figure 7. In all samples, the Ni 2p_1/2_ peak centers at a binding energy of 871.4 eV, and the Ni 2p_3/2_ peak at 854.0 eV. Considering that the binding energy of Ni^3+^ and Ni^2+^ at 2p_3/2_ position is 855.8 and 853.8 eV, a large content of Ni^2+^ and a small content of Ni^3+^ exist in the sample based on the precursor synthesized at 220 °C. On the other hand, in other samples, almost only Ni^2+^ can be detected, corresponding to the peak strength ratio of *I_(003)_*/*I_(104)_* in Table 2. That is to say, in LNCMO prepared at 220 °C, a severe cation mixing phenomenon occurred during the preparation of LNCMO related to the presence of two coexisting Ni^3+^ and Ni^2+^ ions.

For the Co element, the Co 2p_3/2_ and Co 2p_1/2_ peaks were detected at the binding energy of 779.1 and 795.4 eV, respectively, indicating that all of the Co elements in the sample exist in Co^3+^. The binding energy of the Mn element is located at 641.2 and 652.7 eV, respectively, indicating that the Mn element exists at Mn^4+^. The above analysis confirms that the valence state of the Ni element is directly related to the cation mixing degree, and thus the electrochemical performance of the samples. Therefore, to reduce the cation mixing phenomenon in the preparation process of high-nickel ternary LNCMO materials, careful tailoring of the valence state of the nickel element is necessary.

### 3.3. Electrochemical Performances of LNCMO Cathode Materials

#### 3.3.1. Charge/Discharge Cycling Stability

To evaluate the effect of precursor hydrothermal temperature on the electrochemical cycling performance of Ni-rich LNCMO cathode materials, the electrodes were tested between 3.0 and 4.3 V. Figure 8a, b shows the 1st and 5th cycle profiles. It is noted that the initial charge/discharge capacity of the cathode obtained from the precursor of 180 °C is the lowest, which is consistent with its relatively poor crystallinity. The cathode at a hydrothermal temperature of 200 °C exhibits the highest initial charge specific capacity (227 mAh g^−1^), while the cathode using a precursor of 220 °C shows the highest discharge-specific capacity (141 mAh g^−1^). On the other hand, the Coulombic efficiency is generally low (60–70%) at the 1st cycle, but is significantly improved by ~20% after the 5th cycles, implying that the solid electrolyte interphase (SEI) film is probably formed on the positive electrode surface.

The cathode materials were then further subjected to more charge/discharge cycles, as presented in Figure 9. Under 0.1C, as shown in Figure 9a, the specific capacity decreases at a higher rate for several initial cycles; then gradually decreases at a lower rate until 50 cycles. After 50 charge/discharge cycles, the specific discharge capacity is 92.6, 99.5, 118.0 and 107.1 mAh g^−1^, respectively, in the LNCMO obtained using precursors of 180, 200, 220 and 240 °C. The corresponding capacity retention rate is 70.77, 70.21, 81.94 and 75.35%, respectively. In particular, the sample at a hydrothermal temperature of 220 °C shows the best cycle performance since it has the smallest particle size, as shown in Figure 5e,f. Usually, smaller particle size implies a larger specific surface area and thus larger contact area of the positive electrode material with electrolytes. Consequently, the lithium-ion diffusion path may be shorter, resulting in better charge/discharge cycle performance. In addition, cycle stability has a close relationship to structural stability. Due to the phenomenon of cation mixing in high-nickel ternary cathode materials, structural transition (from a layered to a non-electrochemically active rock salt phase structure) occurs during charge and discharge processes, thereby impeding lithium-ion migration, increasing material resistance and thus reducing the cycle performance of the cathode material. The reason for the best cycle stability of the sample at 220 °C is that the precursor synthesized at this temperature is highly crystalline and has few lattice defects. As a result, it exhibits enhanced structural stability and thus cyclic stability.

Figure 9b shows the rate performance at current densities of 0.1C–2C, with 5 cycles at each current density. It can be seen that the sample based on the precursor of 220 °C has an average discharge specific capacity of 152.0, 135.5, 113.0, 86.6 and 57.1 mAh g^−1^ at the rate of 0.1C, 0.25C, 0.5C, 1C and 2C, respectively. During charging or discharging at low or high currents, the sample based on the precursor of 220 °C exhibits the highest capacity compared to the other three samples. This means that the sample obtained from the precursor of 220 °C has the shortest lithium-ion diffusion path due to having the smallest particle size.

#### 3.3.2. Energy Density

Energy density is one of the most important properties of lithium-ion batteries [33,34], and can be defined as discharge specific capacity × discharge median potential/weight. The decrease in the discharge potential of the positive electrode material directly leads to the attenuation of the energy density of the material. Therefore, in order to obtain stable energy density, the potential stability of the electrode material is equally important. Figure 10 shows the change in the midpoint potential of the four samples during charge and discharge cycles. It can be seen that the discharge midpoint potential of the sample with a hydrothermal temperature of 220 °C is the most stable, with the discharge midpoint potential decreasing by only 0.0245 V (i.e., attenuation rate 0.65%) after 50 cycles. Additionally, the charge midpoint potential can characterize the degree of electrochemical polarization of the material. After 50 cycles, the midpoint potential growth rates of samples at 180–240 °C were 2.56, 1.76, 0.15 and 1.99%, respectively, indicating that samples prepared at 220 °C can effectively inhibit the electrochemical polarization of materials.

Supposing that the battery volume and mass remain unchanged, and the energy consumption of electric vehicles keeps the same, the higher the energy density, the greater the driving distance of an electric vehicle. Figure 11 shows the discharge energy density for samples prepared using different precursors. It can be clearly seen that the sample based on the precursor of 220 °C has the highest energy density. Furthermore, except for the fast decay of energy density in the first few cycles, energy density remains stable throughout the subsequent cycles. The decay of energy density from the 4th to the 50th cycles is only 12.13%. For the samples with hydrothermal temperatures of 180 °C and 240 °C, the initial energy densities are similar to that of the sample of 220 °C, but decrease rapidly with an increasing number of charge and discharge cycles, leading to the reduced service life of the battery. As for the sample based on the precursor of 200 °C, the energy density stability keeps 82% of the initial value after 50 cycles, but its overall energy density is obviously lower than the sample with a hydrothermal temperature of 220 °C. Based on the above analysis, the sample with a hydrothermal temperature of 220 °C not only has the highest specific capacity, the best cycling stability and the best rate performance, but also the highest and most stable energy density.

#### 3.3.3. Impedance and Morphology of LNCMO after Cycle

To analyze the reasons for the above electrochemical performances of the four samples, a kinetic analysis was conducted after multiple cycles. Figure 12 shows the electrochemical impedance spectra (EIS) of samples. The measured state is the charging condition in the 51st cycle, with a charging voltage of 3.6–3.7 V. EIS curves are composed of a half arc in a high-frequency region and another arc in middle- and low-frequency regions. The high-frequency region corresponds to the diffusion impedance *R_f_* of Li^+^ in the SEI film, and the left intersection point of the half circle and axis to the Ohmic impedance *Rs* inside the battery. The arc in the middle- and low-frequency regions is related to the charge transfer process, whose radius represents the charge transfer impedance. The main part of the Nyquist diagram is the arc part in the middle- and low-frequency regions, indicating the charge transfer process of the main electrochemical process. The charge transfer impedance of the samples with hydrothermal temperatures of 220 °C is much smaller than that of the other three samples, since the sample of 220 °C has a relatively small particle size. In addition, the sample with a hydrothermal temperature of 180 °C has the highest *R_f_*, which is probably related to the thicker SEI film covering its surface, which is confirmed in the SEM images of the electrode after cycling (Figure 13).

Figure 13 shows the surface morphologies of the cathode materials after charge and discharged cycles for 50 times. It is noted that the surface of the sample with a hydrothermal temperature of 180 °C has a thin, translucent SEI film (Figure 13a). Usually, the formation of SEI film is harmful to the charge/discharge efficiency of electrode material. In addition, the SEI film formed on the surface of the sample is unstable and prone to crack (see Figure 13a). The unstable and cracked SEI film may react with the electrolyte, resulting in poor cycle stability, heat release by the side reaction and thus uncontrollable internal temperature rise in the battery. Consequently, the safety performance of the battery cannot be guaranteed. On the other hand, Figure 13b–d shows that the positive electrode material based on the precursors of 200 °C, 220 °C and 240 °C do not induce SEI films after cycling for 50 times. In these three samples, the sample of 220 °C maintains the best morphology and particle integrity compared to the samples of 200 °C and 240 °C, which is consistent to the best cycling stability shown in Figure 9 and stable charge midpoint potential shown in Figure 10.

Hydrothermal temperature below 180 °C is not conducive to Ni^2+^ precipitation, resulting in less nickel content in the precursor and low crystallinity. In contrast, the precursors prepared at 180~240 °C have better crystallinity. The precursors synthesized by the hydrothermal method show a pine cone shape, which exhibit a large specific surface area and mesoporous structure and thus help in enhancing the uniform distribution of lithium salts. The cathode materials prepared after calcination maintain the morphology of the precursor and have good crystallinity. Li et al. [35] showed that the LiNi_0.8_Co_0.1_Mn_0.1_O_2_ obtained by co-precipitation and calcination is spherical with a low specific surface area with mesoporous structure, which is not conducive to the contact between electrolyte and cathode material. Guo et al. [36] believe that cationic mixing can limit the migration of lithium-ions in the layered materials, which deteriorate the rate performance and charge/discharge capacity of the materials. In our work, the increase of Ni^3+^ content is realized by regulating the hydrothermal temperature, which helps in reducing the degree of cation mixing. Consequently, the cathode material prepared based on the precursors synthesized at a hydrothermal temperature of 220 °C exhibited the best electrochemical cycling performance with a specific capacity retention of 81.94% after charge/discharge cycling for 50 times. In contrast, the capacity retention rate of Na-doped Ni-rich LiNi_0.8_Co_0.1_Mn_0.1_O_2_ prepared by Hwang et al. [37] is only 72.7%. Of course, compared with the coprecipitation method, the hydrothermal method has relatively poor repeatability and low cost-effectiveness, which limits its large-scale production potential. Obviously, more extensive research works are required to reduce the production cost and improve the controllability of the preparation process by tailoring the equipment and experimental process.

## 4. Conclusions

The effect of hydrothermal temperature on the structure, morphology, and cycling performance of Ni-rich LiNi_0.8_Mn_0.1_Co_0.1_O_2_ cathode materials was systematically investigated. The results demonstrate that the cathode material prepared based on the precursors synthesized at a hydrothermal temperature of 220 °C exhibited the best electrochemical cycling performance with specific capacity retention of 81.94% after charge/discharge cycling for 50 times. The enhanced properties were attributed to the small crystalline grain size, large specific surface area, and thus a large contact area with the electrolyte. On the other hand, the higher crystallinity of the precursor synthesized at 220 °C that led to the high crystallinity, superior structural stability, stable charge/discharge medium potential, and absence of SEI film formation during charge/discharge cycles, also contributed to the stable electrochemical characters.

## Figures and Tables

**Figure 1 materials-16-05398-f001:**
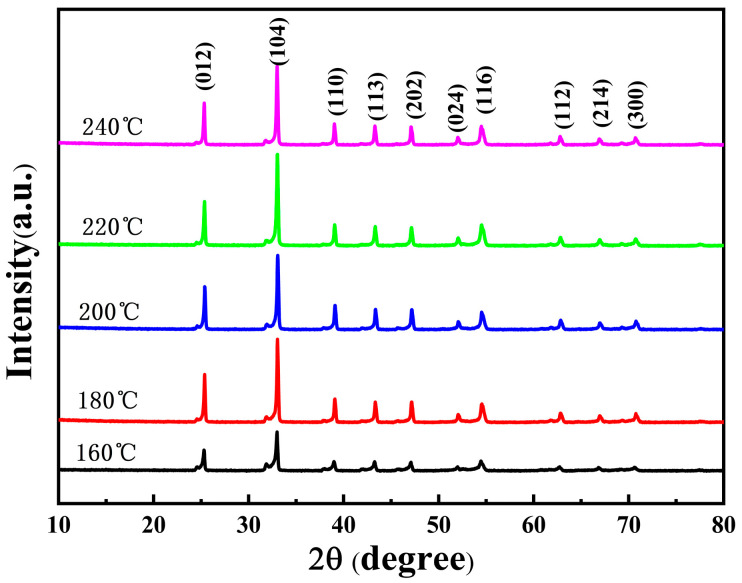
X-ray diffraction (XRD) patterns of NCMO precursors synthesized at hydrothermal temperatures of 160, 180, 200, 220 and 240 °C.

**Figure 2 materials-16-05398-f002:**
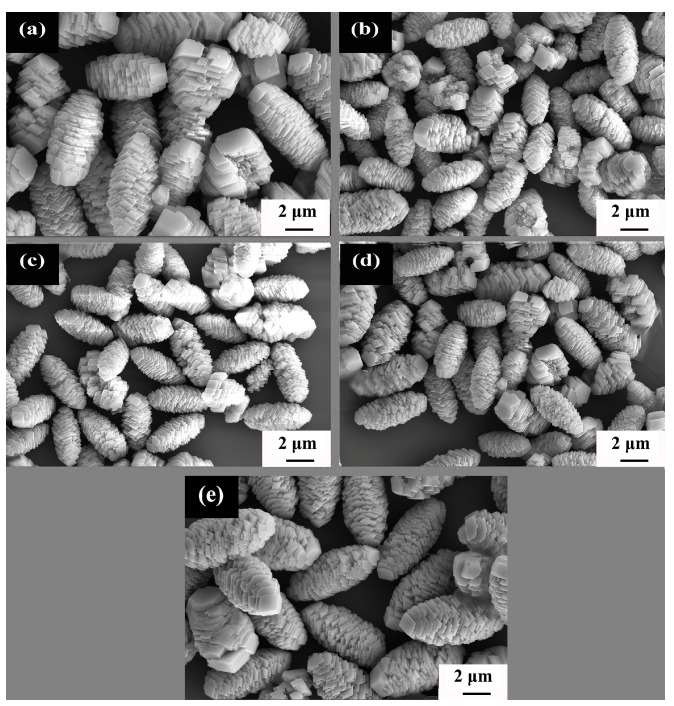
SEM images of NCMO precursor crystals hydrothermally synthesized at different temperatures. (**a**) 160 °C, (**b**) 180 °C, (**c**) 200 °C, (**d**) 220 °C, (**e**) 240 °C.

**Figure 3 materials-16-05398-f003:**
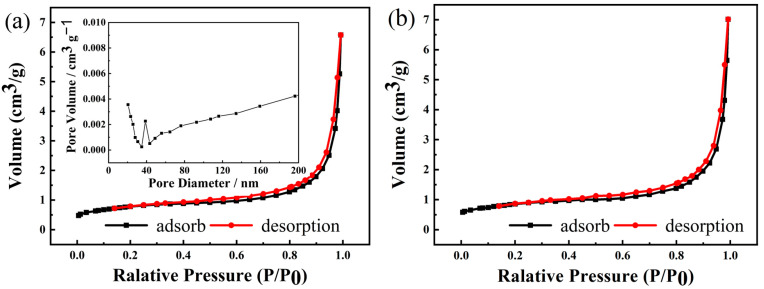
N_2_ adsorption and desorption curve of the NCMO precursor synthesized at 180 °C (**a**) and 220 °C (**b**) (inset is pore size distribution curve of the sample).

**Figure 4 materials-16-05398-f004:**
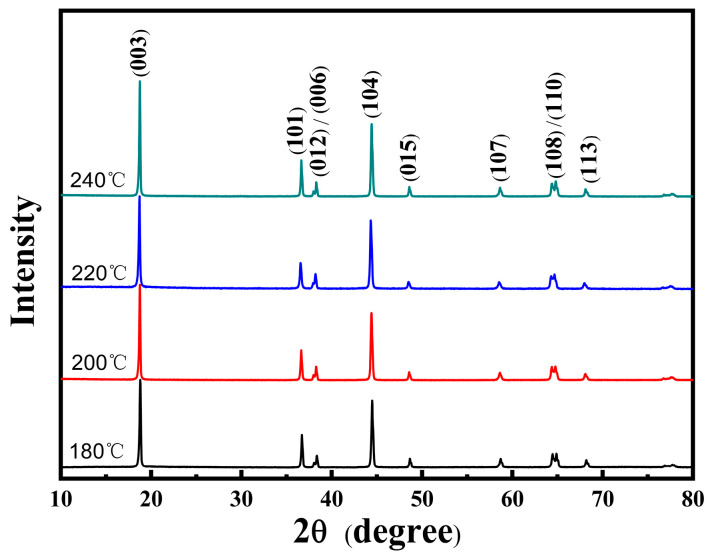
XRD patterns of the LNCMO cathode samples using precursors synthesized at the different hydrothermal temperatures of 180, 200, 220 and 240 °C.

**Figure 5 materials-16-05398-f005:**
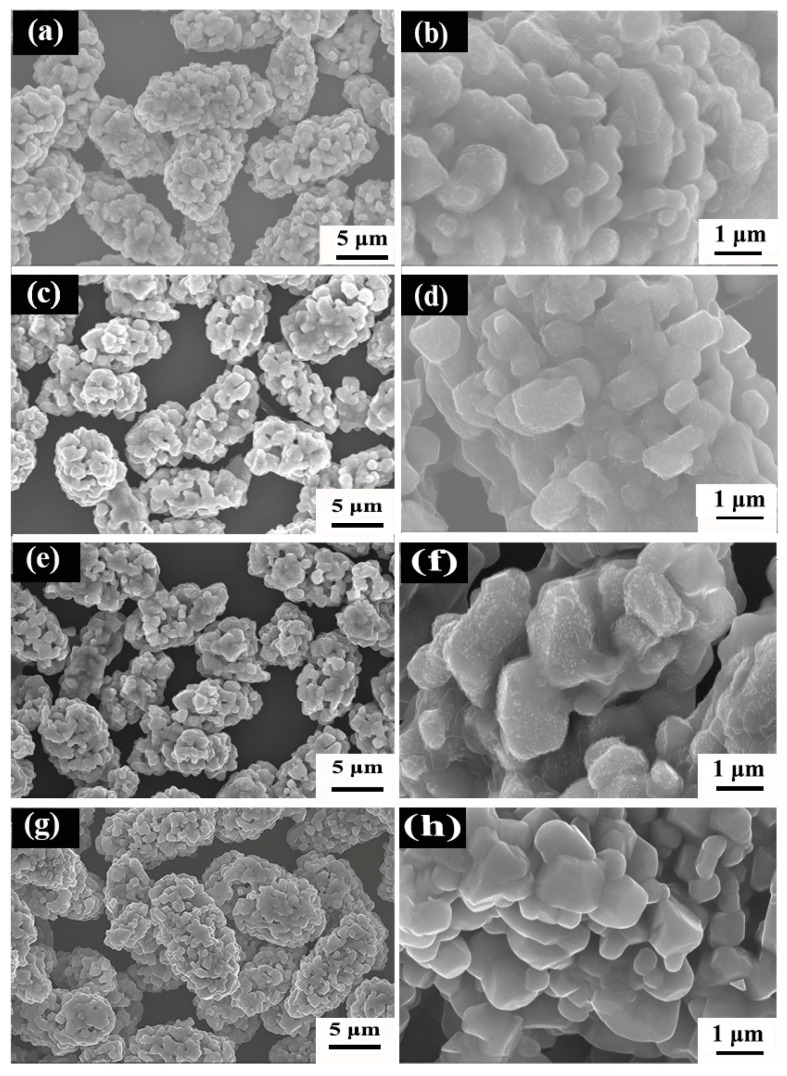
SEM images of LNCMO cathode materials after calcination of precursors obtained at different hydrothermal temperatures: (**a**,**b**) 180 °C, (**c**,**d**) 200 °C, (**e**,**f**) 220 °C and (**g**,**h**) 240 °C.

**Figure 6 materials-16-05398-f006:**
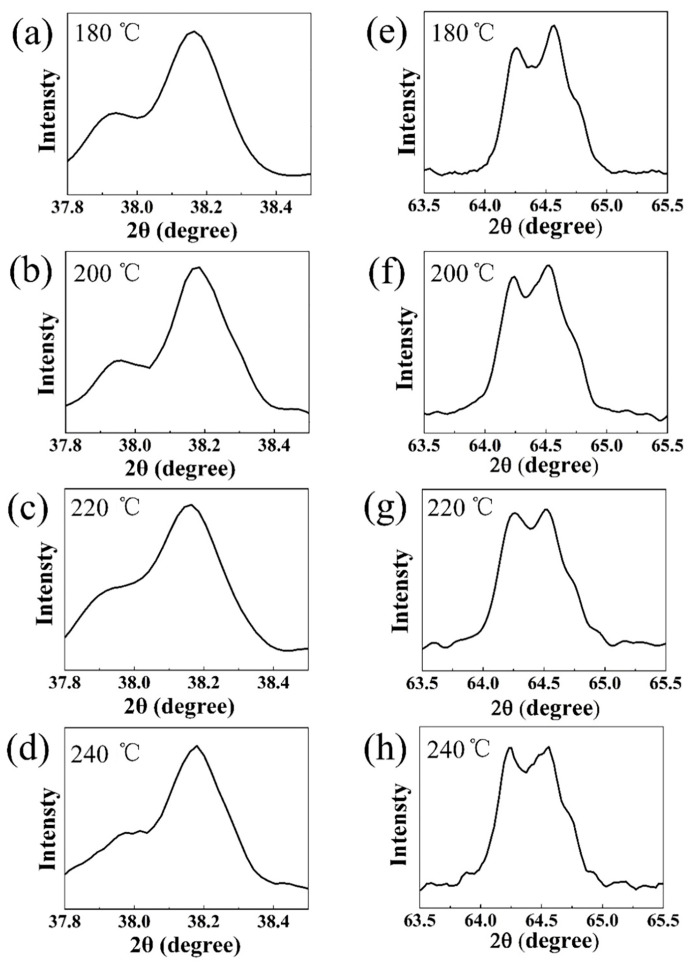
Enlarged view of XRD peaks in LNCMO prepared using precursors obtained at different hydrothermal temperatures. (**a**–**d**) (012)/(006) peaks, (**e**–**h**) (108)/(110) peaks.

**Figure 7 materials-16-05398-f007:**
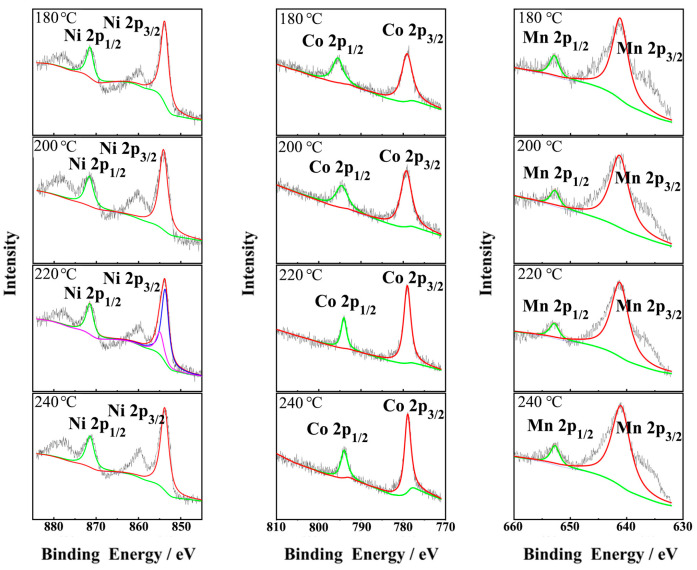
XPS spectra of LNCMO using precursors prepared at different temperatures.

**Figure 8 materials-16-05398-f008:**
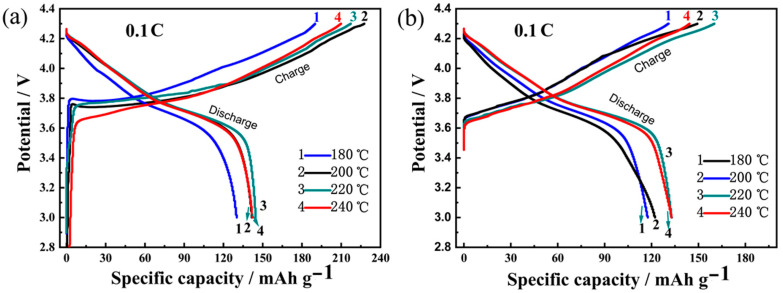
Charge/discharge cycle profiles of LNCMO cathode materials prepared from precursors of 180, 200, 220 and 240 °C. (**a**) 1st cycle, (**b**) 5th cycle.

**Figure 9 materials-16-05398-f009:**
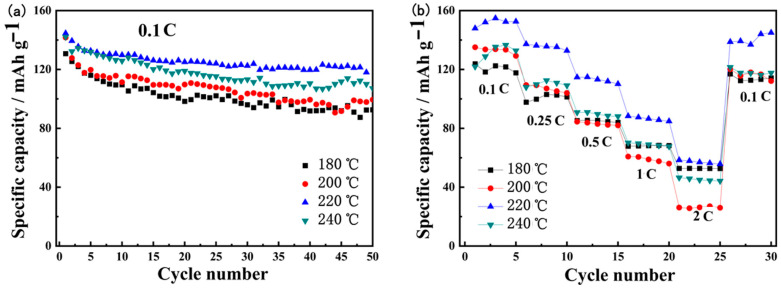
Charge/discharge cycling specific capacity at current density of (**a**) 0.1 C and (**b**) 0.1–2 C in LNCMO materials prepared using precursors of 180, 200, 220 and 240 °C.

**Figure 10 materials-16-05398-f010:**
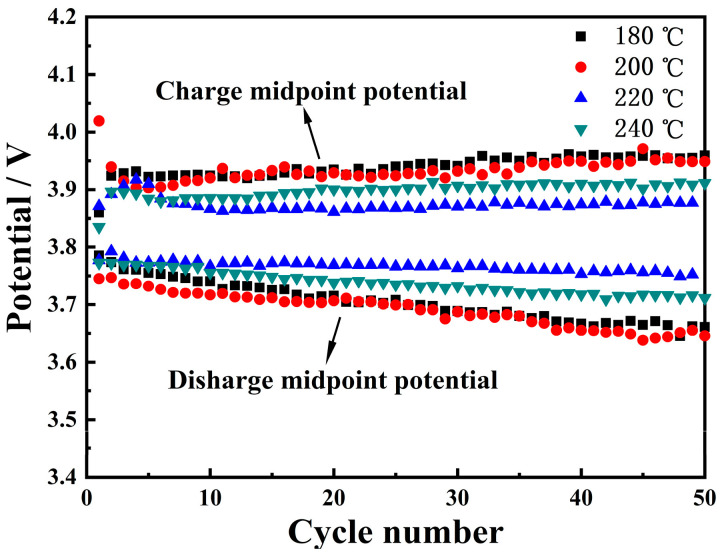
Midpoint potential of LNCMO materials after calcination of precursors of 180, 200, 220 and 240 °C.

**Figure 11 materials-16-05398-f011:**
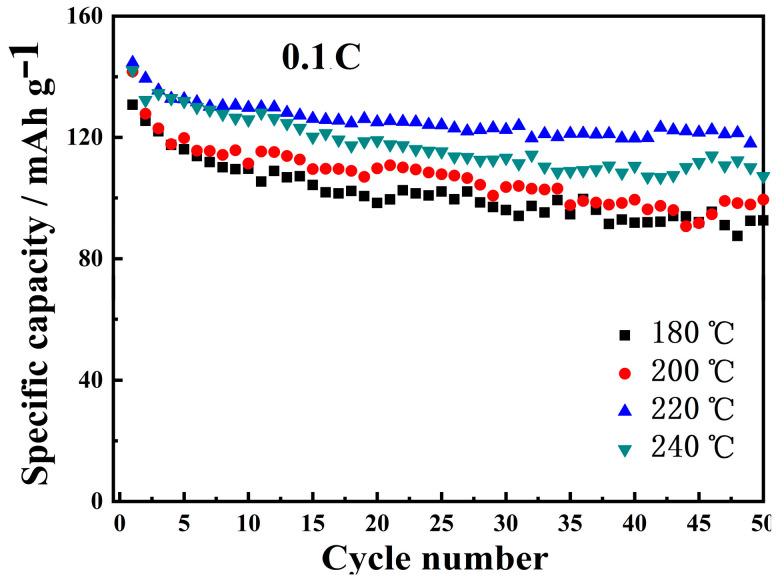
Energy density of LNCMO prepared using precursors of 180, 200, 220 and 240 °C.

**Figure 12 materials-16-05398-f012:**
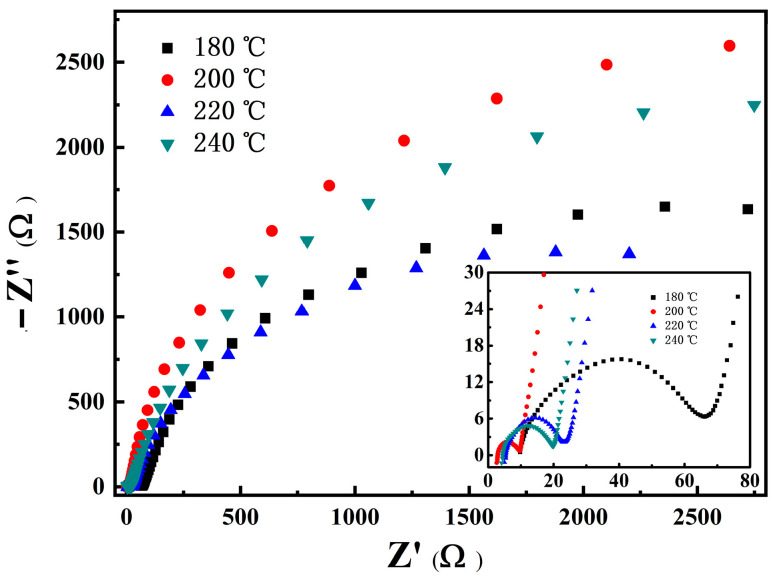
Nyquist curves of LNCMO prepared using precursors of 180, 200, 220 and 240 °C.

**Figure 13 materials-16-05398-f013:**
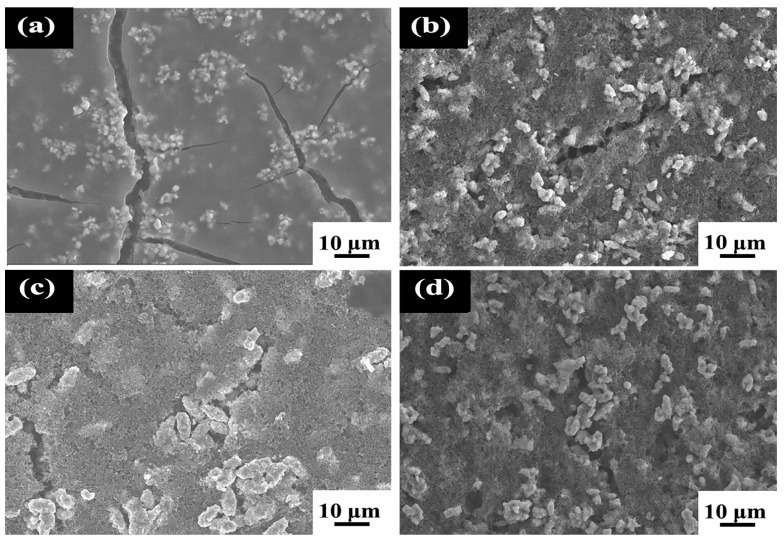
SEM images of LNCMO after charge and discharge cycles for 50 times. The precursors are synthesized at temperatures of (**a**) 180 °C, (**b**) 200 °C, (**c**) 220 °C, and (**d**) 240 °C.

**Table 1 materials-16-05398-t001:** Composition of NCMO precursors synthesized at different temperatures.

Temperature (°C)	Ni (mmol/L)	Co (mmol/L)	Mn (mmol/L)	Ni:Co:Mn Ratio
160	1.341	0.268	0.298	4.50:0.90:1
180	1.226	0.186	0.144	8.51:1.22:1
200	1.195	0.175	0.144	8.30:1.20:1
220	1.285	0.187	0.155	8.29:1.21:1
240	1.425	0.182	0.178	8.01:1.02:1

**Table 2 materials-16-05398-t002:** Crystallographic parameters (*a*, *b* and *c*) and peak intensity ratio of (003) and (104) planes (*I_(003)_*/*I_(104)_*) in LNCMO samples from precursors synthesized at different temperatures.

Temperature (°C)	*a* (Å)	*b* (Å)	*c* (Å)	*c*/*a*	*I_(003)_*/*I_(104)_*
180	2.875	2.875	14.217	4.945	1.20
200	2.874	2.874	14.200	4.941	1.58
220	2.873	2.873	14.207	4.945	1.94
240	2.875	2.875	14.207	4.942	1.50

## Data Availability

The data are available from the corresponding author upon reasonable request.

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
