# Peer review of "Enhancing the Electrochemical Stability of LiNi0.8Co0.1Mn0.1O2 Compounds for Lithium-Ion Batteries via Tailoring Precursors Synthesis Temperatures"

_materials, 2023, doi:10.3390/ma16155398_

Round 1

Reviewer 1 Report

Figure 1 is blurred, it should be replaced by a clearer picture. 

How did the authors come to the reaction equation during hydrothermal process? Are there any literature data or experimental measurements to support them? If not, it should be skipped.

How did the lattice parameters were calculated? It would also be important to analyze the crystallite size of the powders in line with particle size.

There is a typo (change/discharge) in several places in the text (also in the abstract).

Reviewer 2 Report

Review on the manuscript

Enhancing Electrochemical Stability of LiNi0.8Co0.1Mn0.1O2
Compounds for Lithium-ion Batteries via Tailoring Precursors
Synthesis Temperatures

In this work, the authors have illustrated the preparation of LiNi0.8Co0.1Mn0.1O2 (LNCMO) cathode materials for lithium-on batteries (LIBs) by hydrothermal synthesis of precursors and high temperature calcination. Then, the results showed that the cathode material prepared using the precursor synthesized at 15 a hydrothermal temperature of 220°C exhibited the best charge/discharge cycle stability….The paper is well-written and easy for the reader to follow.

In short, I feel the subject of this article is interesting, applicable  and matches the scope of the journal but needs some major and minor revisions.

Details:

Some major revisions

1. In the introduction part the authors need to cite the recent Lithium-ion Batteries published articles. Beside, in the introduction part, the research process should be described in detail. Is there any other literature supporting these conclusions?

2. The Results and discussion has started with a figure. I think it is better to start this section with some explanations.

3. The quality of some Figures is inferior and needs to be enhanced. Please improve the quality of all Schemes and Figures and Graphs in your manuscript to be more clear and legible.

4. Some of Equations are not numbered. Please check it in whole the text.

5. Results and Discussion need to be separated separately. The two are indispensable and cannot be combined, please modify.

6. The discussion part of this article needs to be rewritten. The discussion should include: the main findings of this experiment, the comparison of the research results with those reported in the literature, the advantages and limitations of the research, and the clinical practice, Implications for future research, etc.;

Some minor revisions as follows:

1. Line 11, lithium-on batteries should be lithium-ion batteries.

2. Line 16, I think you mean charge/discharge.

3. Line 42, I think you mean charge/discharge.

4. Line 54, Then, the solution…Besides, please check all punctuations in the manuscript.

5. Please increase the quality of Figure1.

6. Line 117, According to Equation (1)-(3), … should be Equations.

7. Please increase the quality of Figures 2, 4, 5,6,7.

8. Line 259, …better change/discharge cycle performance should be charge/discharge.

9. There are some syntax errors and grammatical problems, and the paper should be revised carefully. It is suggested that the author read through the article for several rounds to make corrections.

There are some syntax errors and grammatical problems, and the paper should be revised carefully. It is suggested that the author read through the article for several rounds to make corrections.

Reviewer 3 Report

The article is devoted to an important topic - the synthesis and study of the properties of Ni-rich cathode materials. The article studies NMC811 samples obtained on the basis of NMCO synthesized at different temperatures by the hydrothermal synthesis method.

The structure of the article complies with the requirements of the journal. However, there are errors and typos in the text. The authors should pay attention to the text and carefully rework it.

1. Introduction is too short and does not contain enough information on the topic of the article. In addition, there are errors: on line 37, the authors refer to articles on the hydrothermal synthesis of LNCMO, but these articles are devoted to the production of Li[Li1/3-2x/3NixMn2/3-x/3]O2 and Li0.93[Li0.21Co0.28Mn0 .51]O2.

2. There are typos on lines 16 and 42: change instead of charge

3. The quality of the drawings â„–1, 2, 4-9, 13 is poor. 

4. It is not clear why a different ratio of precursors was used in the synthesis of NCMO. The authors need to clarify this point.

5. Why does the ratio of N, Mn and Co in NCMO change with increasing temperature?

6. The article indicates the specific surface area of the NCMO sample obtained at 180C. The specific surface area values for the rest of the samples should be added. The article also mentions characteristics such as particle size and specific surface area for LNCMO samples, but the values are not specified anywhere. You should add values to the article.

7. In the paragraph on lines 146-153 (first paragraph on page 5), sentences 3 and 4 are not logically related to each other. The authors write about the features of the pine cone-like structure, and then explain why the sample obtained at 160C is not used in further experiments, although it has the same structure. You should rewrite the paragraph or add clarifications. 

8. How can you explain why a sample synthesized at 220C has maximum crystallinity?

9. On lines 163-164 there is a sentence stating that Li/N cation mixing degree is not very serious. What dose it mean: "not very serious"? It may be worth using another term that is more appropriate in style to a scientific article.

10. In Figure 8a, the charge curves are incorrectly numbered. This should be corrected.

11. The title of clause 3.3.3 Mechanisms for Enhanced Cycling Stability in the Sample of 220°C does not match the content. The name should be changed or more attention should be paid to the cycling stabilization mechanism.

Round 2

Reviewer 2 Report

Dear Editor,

Concerning the responses of authors through their revised manuscript, I am satisfied with the changes that they have made and I thank them for the extra work they have put in. Now, it is a pleasure to accept the manuscript entitled "Enhancing Electrochemical Stability of LiNi0.8Co0.1Mn0.1O2 Compounds for Lithium-ion Batteries via Tailoring Precursors Synthesis Temperatures". 

Dear Editor,

Concerning the Quality of English,  the article of "Enhancing Electrochemical Stability of LiNi0.8Co0.1Mn0.1OCompounds for Lithium-ion Batteries via Tailoring Precursors Synthesis Temperatures" now can be published in this form, only Minor editing of English language required